# Atmospheric impact of 2-methylpentanal emissions: Kinetics, photochemistry, and formation of secondary pollutants

María Asensio[1,2], Sergio Blázquez[1,2,#], María Antiñolo[1,2], José Albaladejo[1,2], Elena Jiménez[1,2,*]

[1]Instituto de Investigación en Combustión y Contaminación Atmosférica, Universidad de Castilla-La Mancha, Camino de Moledores s/n, Ciudad Real, 13071, Spain.
[2]Departamento de Química Física, Universidad de Castilla-La Mancha, Avda. Camilo José Cela 1B, Ciudad Real, 13071, Spain.

* *Correspondence to*: Elena Jiménez (elena.jimenez@uclm.es)

---

[#] Current address. Escuela Técnica Superior de Ingenieros Industriales, Departamento de Química Física, Universidad de Castilla-La Mancha, Avda. España s/n, Albacete, 02071, Spain.

**Abstract.** The tropospheric fate of 2-methylpentanal (2MP) has been investigated in this work. First, the photochemistry of 2MP under simulated solar conditions was investigated by determining the UV absorption cross sections (220-360 nm) and the effective photolysis quantum yield in the UV solar actinic region ($\lambda > 290$ nm). The photolysis rate coefficient in that region was estimated using a radiative transfer model. Photolysis products were identified by Fourier Transform Infrared (FTIR) spectroscopy. Secondly, a kinetic study of the Cl- and OH- reactions of 2MP was also performed at 298 K and as a function of temperature (263-353 K), respectively. For the Cl-reaction, a relative kinetic method was used in a smog chamber coupled to FTIR spectroscopy, whereas for the OH-reaction, the Pulsed Laser Photolysis with Laser Induced Fluorescence (PLP-LIF) technique was employed. The estimated lifetime of 2MP depends on the location, the season, and the time of the day. Under mild-strong irradiation conditions, UV photolysis of 2MP may compete with its OH-reaction in a mid-latitude inland urban atmosphere, while Cl reaction dominates in mid-latitude coastal urban areas at dawn. Finally, the gaseous product distribution of the Cl- and OH-reactions was measured in a smog chamber as well as the formation of secondary organic aerosols (SOAs) SOAs in the Cl-reaction and its size distribution (diameter between 5.6 and 560 nm). The implications on air quality are discussed based on the observed products.

## 1 Introduction

The World Health Organization (WHO) estimates that air pollution is responsible of around 7 million deaths worldwide. One of the damaging air pollutants is *particulate matter* (PM). Concretely, fine and ultrafine particles (UFPs), with diameters smaller than 2.5 and 0.1 μm, respectively, are extremely dangerous since they penetrate deep into the lungs and enter the bloodstream. UFPs and other secondary pollutants are formed in the gas-phase reactions of volatile organic compounds (VOCs), such as aldehydes, with oxidants such as hydroxyl (OH) radicals or chlorine (Cl) atoms. In the atmosphere, OH radicals are ubiquitous, while Cl is mainly present in marine or coastal atmospheres since it is formed from reactions of sea-salt aerosols (Atkinson et al., 1995; Wennberg et al., 2018; Rodríguez et al., 2012; Atkinson and Arey, 2003; Antiñolo et al., 2019; Antiñolo et al., 2020). Globally, ceramic industries and coal-fired power plants are relevant sources of $Cl_2$ inlands (Galán et al., 2002; Sarwar and Bhave, 2007; Deng et al., 2014).

Saturated aldehydes, like formaldehyde, acetaldehyde, pentanal, and hexanal, are emitted into the atmosphere (Clarisse et al., 2003; Calvert et al., 2011; Villanueva et al., 2022) mainly from primary sources, *e.g.* natural gas from power stations, landfill gas, flaring from offshore, and transport (Calvert et al., 2011). But they can also be formed in situ in the atmosphere as secondary pollutants from reactions of alkenes and alcohols with oxidants (Atkinson et al., 1995; Wennberg et al., 2018; Rodríguez et al., 2012; Atkinson and Arey, 2003). In this work, we focus on the diurnal atmospheric chemistry of 2-methylpentanal (2MP, $CH_3CH_2CH_2CH(CH_3)C(O)H$) and its impact in the troposphere in terms of its lifetime or generated products. This aldehyde has been detected in the atmosphere (Xu et al., 2003) as it can be released to the environment from some foods (Aisala et al., 2019) and from contaminated water in waste streams (Bao et al., 1998), since it is widely used as a flavoring ingredient and as an intermediate in the synthesis of dyes, resins, and pharmaceuticals (Furia and Bellanca, 1975). Moreover, 2MP has been detected in ambient air at the foot of Mount Everest (Ciccioli et al., 1993). The former source of 2MP implies that it comes from localized sources but the latter one could imply that 2MP is both long-lived and well-mixed throughout the troposphere or has a specific local source in that remote region. In addition, 2MP has been detected in certain indoor environments from emission of cigarette smoke (Lippmann, 2000).

Once emitted, 2MP can be degraded during daytime by different processes forming secondary pollutants (gases and/or PM) that can have a significant impact on air quality and health. In the atmosphere, these processes include ultraviolet (UV) photolysis by the solar actinic radiation (Reaction R1), reaction with Cl atoms (Reaction R2) and/or reaction with OH radicals (Reaction R3).

$$CH_3CH_2CH_2CH(CH_3)C(O)H + h\nu \ (\lambda \geq 290 \ nm) \rightarrow Products \qquad J \qquad (R1)$$

$$CH_3CH_2CH_2CH(CH_3)C(O)H + Cl \rightarrow Products \qquad k_{Cl} \qquad (R2)$$

$\qquad CH_3CH_2CH_2CH(CH_3)C(O)H + OH \rightarrow Products \qquad k_{OH} \qquad (R3)$

To our knowledge, the photolysis rate coefficient ($J$) of 2MP in the atmosphere and the rate coefficient for Reaction R2, $k_{Cl}$, were not reported up to date in the literature. D'Anna et al. (2001) reported the rate coefficient for the 2MP + OH reaction (Reaction R3), $k_{OH}$, at 298 K and 760 Torr of air. The reaction products of reactions R1-R3 are still unknown.

In this work, the evaluation of the atmospheric fate of 2MP has been carried out under NO$_x$-free conditions, simulating a clean atmosphere. Therefore, the UV photodissociation of 2MP has been investigated at room temperature, determining the UV absorption cross sections ($\sigma_\lambda$) between 220 and 360 nm and the effective photolysis quantum yield ($\Phi_{eff}$) and $J(z,\theta)$ in the actinic region in a Spanish inland city (Ciudad Real) and a mid-latitude coastal urban city (Valencia). The rate coefficients of reactions R2 and R3 have been determined by the relative method using Fourier Transform Infrared (FTIR) spectroscopy and

by Pulsed Laser Photolysis – Laser Induced Fluorescence (PLP-LIF), respectively. The gas-phase products of reactions R1-R3 have been also detected by Gas Chromatography coupled to Mass Spectrometry (GC-MS), FTIR spectroscopy, and Proton Transfer Reaction - Time of Flight – Mass Spectrometry (PTR – ToF – MS). No information on the formation of secondary organic aerosols (SOAs) of reaction R1-R3 have been reported yet. In this work, we study the SOA formation in reaction R2 by using a Fast Mobility Particle Sizer Spectrometer (FMPS). The potential implications of the diurnal degradation processes

of 2MP are discussed in terms of the estimated lifetime in mid-latitude inland and coastal urban atmospheres. The identification of the gaseous products formed in reactions R1-R3 as well as the formation of submicron particles in the reaction R2 provide a better understanding of the tropospheric photochemistry of 2MP and its impact on air quality.

## 2 Experimental section

### 2.1 Atmospheric photodissociation of 2MP

The experimental systems and procedures have been described in detail elsewhere (Blázquez et al., 2020; Asensio et al., 2022), so only a brief description is given here. The UV absorption cross sections of 2MP ($\sigma_\lambda$ in base $e$) were determined by UV absorption spectroscopy (Blázquez et al., 2020) between 220 and 360 nm at (298 ± 1) K. A cylindrical cell with an optical pathlength of 107.15 cm was filled with gas-phase 2MP in the range of 1.02 – 9.65 Torr and irradiated by a deuterium lamp (DT-200, StellarNet). The transmitted radiation was detected by a 2048-pixels CCD camera coupled to a Czerny-Turner

spectrograph (BLACK-Comet model C, StellarNet) with 3 nm spectral resolution. $\sigma_\lambda$ were determined from the slopes of Beer-Lambert's law plots, such as the showed in Figure S1 and as described in the SI.

The experimental set-up used to investigate the photolysis of 2MP by sunlight under atmospheric conditions ((298±1) K and (760±2) of air) has been described elsewhere (Asensio et al., 2022). Briefly, a 20-cm long cylindrical jacketed cell was filled with diluted 2MP in synthetic air (dilution factors, $P_{2MP}/(P_{2MP}+P_{air})$, ranged from 1.07 to 1.73 × 10$^{-3}$). The temperature of the

85 gas mixture in the cell was maintained constant by recirculating water through the jacket from a thermostatic bath (CD-200F, CORIO). The gas mixture was irradiated with a solar simulator (model 11002-2, SunLite$^{TM}$) at $\lambda$ > 290 nm during 30, 60, 90, 120, or 150 min. Average irradiance was (2.335 ± 0.126) Suns. One Sun is the irradiance of the AM1.5G reference solar spectrum (Gueymard et al., 2002). At each photolysis time, the gas mixture (unreacted 2MP and reaction products) was transferred to a 16-L White-type cell, with an optical path length of 71 m, coupled to a FTIR spectrometer (Nicolet Nexus 870,

Thermo Fisher Scientific) (Asensio et al., 2022) with a 2 cm$^{-1}$ resolution. Spectra were recorded between 650 and 4000 cm$^{-1}$ and after the accumulation of 32 interferograms. The selected IR bands for monitoring 2MP were those centered at 1748 cm$^{-1}$ and between 2630 and 3000 cm$^{-1}$.

In some experiments, cyclohexane was added to the gas mixture as a radical-scavenger ([cyclohexane]$_0$/[2MP]$_0$ = 10.5 – 8.8) to evaluate the impact of secondary chemistry. Six of a total of ten experiments were performed adding cyclohexane, finding no difference in the photolysis rate coefficient, $J$, and photoproducts were identified in the absence of cyclohexane. In addition, dark experiments were performed to evaluate the loss of 2MP by heterogeneous reaction onto the reactor walls. The rate coefficient, $k_{heterog}$, was determined to be $1.43 \times 10^{-5}$ s$^{-1}$. The heterogeneous loss process accounts for 39 % of the total 2MP loss. Therefore, the total loss of 2MP after irradiation was corrected with $k_{heterog}$ to obtain $J$ (in s$^{-1}$) from the slope of the plot of $ln([2MP]_0/[2MP]_t)$ versus time according to Eq. (E1).

$$ln([2MP]_0/[2MP]_t) = (k_{heterog}+J) \ t \tag{E1}$$

where the subscripts 0 and $t$ refer to the concentrations of the 2MP at initial time and elapsed time $t$, respectively. The initial concentration of 2MP in the photolysis cell ranged from 1.12 to $6.55 \times 10^{16}$ molecules cm$^{-3}$ (i.e., 455-2663 ppm).

## 2.2 Gas-phase kinetics of the reaction of 2MP with Cl or OH

### 2.2.1 Relative measurements of $k_{Cl}$

The relative rate methodology and the experimental system to determine $k_{Cl}$ at (298±2) K and (760±5) Torr–were already described (Antiñolo et al., 2019; Antiñolo et al., 2020). In this work, isoprene and propene were used as reference compounds which react with Cl ($k_{ref}$) in competition with 2MP. The mixture of 2MP, Cl$_2$, one of the reference compounds, and synthetic air was introduced into the 16-L cell described above and FTIR spectroscopy was used as detection technique to monitor 2MP and the reference compounds (isoprene or propene). IR spectra were recorded every 2 min and the IR bands selected for monitoring 2MP, isoprene, and propene were 2630-3000 cm$^{-1}$, 3100 cm$^{-1}$, and 3050-3100 cm$^{-1}$, respectively. Ranges of initial concentrations were [2MP]$_0$ = (3.7 - 9.4) $\times 10^{14}$ molecules cm$^{-3}$ (15 - 38 ppm), [isoprene]$_0$ = (3.7 - 6.2) $\times 10^{14}$ molecules cm$^{-3}$ (15-25 ppm) or [propene]$_0$ = (4.2 - 6.2) $\times 10^{14}$ molecules cm$^{-3}$ (17 - 25 ppm), and [Cl$_2$]$_0$ = (3.4 - 4.2) $\times 10^{14}$ molecules cm$^{-3}$ (14 - 17 ppm). Cl atoms were generated in situ by photolysis of Cl$_2$ by 3 actinic lamps (Philips Actinic BL TL 40W/10 1SL/25, $\lambda$ = 340-400 nm) surrounding the cell. Both 2MP and the reference compound mainly react with Cl, but they can also be removed by heterogeneous reaction onto the reactor walls, UV photolysis and/or reaction with Cl$_2$. The rate coefficient for these three loss processes of 2MP ($k_{loss}$) and the reference compound ($k_{ref,loss}$) was evaluated prior each experiment (Antiñolo et al., 2019; Antiñolo et al., 2020). As shown in Table S1, 2MP and the reference compounds only exhibited wall losses and no photolysis at the emission wavelengths of the actinic lamps. Only isoprene reacts with Cl$_2$.

Considering all the processes, the integrated rate equation is given by Eq. (E2).

$$\ln\left(\frac{[2MP]_0}{[2MP]_t}\right) - k_{loss}t = \frac{k_{Cl}}{k_{ref}}\left[\ln\left(\frac{[Ref]_0}{[Ref]_t}\right) - k_{ref,loss}t\right] \tag{E2}$$

The total loss of 2MP and the reference compounds in the absence of Cl atoms were on the order of $10^{-5}$ s$^{-1}$ in all cases.

### 2.2.2 Absolute measurements of $k_{OH}(T)$

The Pulsed Laser Photolysis-Laser Induced Fluorescence (PLP-LIF) technique was employed to determine $k_{OH}$ as a function of temperature ($T$ = 263 – 353 K) and total pressure ($P_T$ = 50 – 500 Torr). The experimental set-up was previously described (Martínez et al., 1999; Albaladejo et al., 2002; Jiménez et al., 2005; Antiñolo et al., 2012; Blázquez et al., 2017; Asensio et al., 2022), thereby a brief description is given here. The reactor consisted of a jacketed Pyrex cell (ca. 200 mL) through which a gas mixture formed by He (bath gas, main flow), H$_2$O$_2$/He and diluted 2MP is flown. The mass flow rates employed in this work are summarized in Table S2 of the SI.

The OH radicals were generated in situ by the PLP of gaseous H$_2$O$_2$ at 248 nm, radiation emitted by a KrF excimer laser (Coherent, Excistar 200). The OH radicals generated were subsequently excited at ca. 282 nm by doubling the output radiation of a Rhodamine-6G dye laser (LiopTech, LiopStar) pumped by the second harmonic of a Nd-YAG laser (InnoLas, SpitLight

1200). At 90 degrees from photolysis and excitation lasers, the LIF at ca. 310 nm was collected by a filtered phototube (Thorn EMI, 9813B). At a constant $T$ and $P_T$, the *pseudo*-first order rate coefficient, $k'$, was obtained from the analysis of the LIF intensity decays shown in Figure S2, at several initial concentrations of 2MP, $[2MP]_0 = (0.30–5.91) \times 10^{14}$ molecules cm$^{-3}$. Then, $k_{OH}(T)$ was determined from the slope of the $k'$ versus $[2MP]_0$ plot. See the SI for more details.

## 2.3 Product studies of 2MP reactions with Cl (R2) and OH (R3)

### 2.3.1. Gaseous products

Briefly, two atmospheric simulation chambers were used to detect and identify the products generated in reactions R2 and R3 at $(759 \pm 3)$ Torr of air and $(298 \pm 2)$ K: the 16-L cell described above coupled to the FTIR spectrometer (only used for the Cl-reaction) (Ballesteros et al., 2017; Antiñolo et al., 2019), and a 264-L chamber (Antiñolo et al., 2020) from which the sample is injected to a GC – MS (Thermo Electron, models Trace GC Ultra and DSQ II) (only used for the Cl reaction) (Asensio et al., 2022) or a PTR – ToF – MS (PTR TOF 4000, Ionicon). The 264-L chamber was surrounded by 8 actinic lamps (Philips Actinic BL TL 40W/10 1SL/25, $\lambda = 340\text{-}400$ nm) to generate Cl atoms by Cl$_2$ photolysis and 4 germicidal lamps (Philips TUV 36W SLV/6, $\lambda = 254$ nm) to generate OH radicals by H$_2$O$_2$ photolysis. In all cases, preliminary tests were carried out in the dark to check if products were generated in the potential reaction of 2MP with the oxidant precursor. In addition, in the absence of oxidant precursor, the formation of products during the UV light exposure of 2MP was also checked. For the product study of the OH-reaction, 2MP/H$_2$O$_2$/air mixtures were irradiated for 70 min by the germicidal lamps, whereas in the Cl atoms experiments, 2MP/Cl$_2$/air mixtures were irradiated with the actinic lamps for 60 min. IR spectra, chromatograms and mass spectra were recorded every 2 min, 15 min and 20 s, respectively.

The GC-MS was equipped with a BPX35 column (30 m × 0.25 mm ID × 0.25 µm, SGE Analytical Science), and the solid-phase microextraction technique was used as sampling method with a 50/30 µm divinylbenzene/carboxen/polydimethylsiloxane (DVB/CAR/PDMS) fiber (Supelco) exposed during 10 min to the gas mixture in the chamber. In these experiments, ranges for the initial concentrations were: $[2MP]_0 = (1.5 \text{ - } 4.4) \times 10^{14}$ molecules cm$^{-3}$ (6-18 ppm) and $[Cl_2]_0 = (1.1 – 2.5) \times 10^{14}$ molecules cm$^{-3}$ (4-10 ppm). Note that the Limit of Detection (LOD) of the GC-MS is $1 \times 10^{11}$ molecules cm$^{-3}$ (4 ppb).

In the FTIR experiments, initial concentrations were slightly different: $[2MP]_0 = (6.1 – 8.5) \times 10^{14}$ molecules cm$^{-3}$ (25-35 ppm) and $[Cl_2]_0 = (1.6 – 5.4) \times 10^{14}$ molecules cm$^{-3}$ (6-22 ppm) (LOD= $8 \times 10^{11}$ molecules cm$^{-3}$ (32 ppb)), while in the PTR-ToF-MS experiments (LOD= $1.2 \times 10^8$ molecules cm$^{-3}$ (5 ppt)), $[2MP]_0 = (1.3 – 2.3) \times 10^{13}$ molecules cm$^{-3}$ (0.5-0.9 ppm) and $[Cl_2]_0 = (1.1 – 1.4) \times 10^{13}$ molecules cm$^{-3}$ (0.4-0.6 ppm), which were lower than those used in the experiments using GC-MS and FTIR due to the higher sensitivity of the PTR-ToF-MS. For the OH-reaction monitored by PTR-ToF-MS, $[H_2O_2]_0 = 3.8 \times 10^{14}$ molecules cm$^{-3}$ (15 ppm). The mass spectra of the reactive gas mixture were recorded with a time resolution of 20 s and the detected mass range was set between 29 and 390.86 amu working at $E/N = 138$ Td and $V_{drift} = 650$V. Prior the *in situ* measurements, a calibration of the PTR-ToF-MS was made between 0.3 and $2.9 \times 10^{13}$ molecules cm$^{-3}$ (122-1179 ppb).

Regarding the oxidant concentrations, [Cl] and [OH] were estimated from the first order loss rate of 2MP measured by FTIR and PTR-ToF-MS. From FTIR experiments, [Cl] was estimated to range between 2.0 and $3.1 \times 10^6$ atoms cm$^{-3}$ (0.08-0.13 ppt), while in the PTR-ToF-MS experiments the estimated [Cl] was found to vary between 1.0 and $2.2 \times 10^7$ atoms cm$^{-3}$ (0.4-0.9 ppt) and [OH]=$4.8 \times 10^5$ radicals cm$^{-3}$ (0.02 ppt).

**2.3.2. Secondary organic aerosols**

To detect and quantify the SOA formation in the Cl-reaction, the 264-L chamber and the 16-L cell were connected in series, as described previously (Antiñolo et al., 2020). SOA experiments were performed in the absence of the reference compounds used in the relative kinetic study (isoprene or propene) to avoid any interference from their degradation initiated by Cl atoms. No seed was added to the gas mixture either. The concentration of 2MP was monitored by the FTIR spectrometer every 2 min, whereas the formed SOAs were monitored by a Fast Mobility Particle Sizer (FMPS) spectrometer (TSI 3091) every 1 min. Initial concentrations were $8.3 \times 10^{14}$ molecules $cm^{-3}$ (34 ppm) of 2MP and $7.8 \times 10^{14}$ molecules $cm^{-3}$ (32 ppm) of $Cl_2$ ([Cl] was estimated to be $8.0 \times 10^5$ atoms $cm^{-3}$ (0.03 ppt). The total timescale of the experiment was 90 min. In the first 15 min, the $Cl_2$/2MP/air mixture was not irradiated to monitor the dark losses of gaseous 2MP or its reaction with $Cl_2$. After that, the lamps were turned on and the Cl reaction started, monitoring 2MP loss and particles during 60 min. Finally, the lights were switched off to evaluate the loss of the SOA formed due to the walls or other dark processes during 15 min.

**3 Results and discussion**

**3.1 Ultraviolet photochemistry of 2MP: Absorption cross sections, photolysis quantum yield, and products**

The Beer-Lambert's law was used to obtain the UV absorption cross sections in the range 220-360 nm at room temperature, as explained in the SI, from nine UV spectra corresponding to nine [2MP]. This series of measurements were duplicated and the average $\sigma_\lambda$ are summarized in Table S3 and depicted in Figure 1 every 1 nm. The absorption maximum was observed at around 296 nm with a peak UV absorption cross section of $\sigma_\lambda = (6.64\pm0.11)\times10^{-20}$ $cm^2$ $molecule^{-1}$. As shown in Figure 1, the solar actinic flux, $F(\lambda,z,\theta)$, overlaps with part of the UV absorption band of 2MP, therefore, UV photolysis in the troposphere of 2MP could be a significant atmospheric removal process.

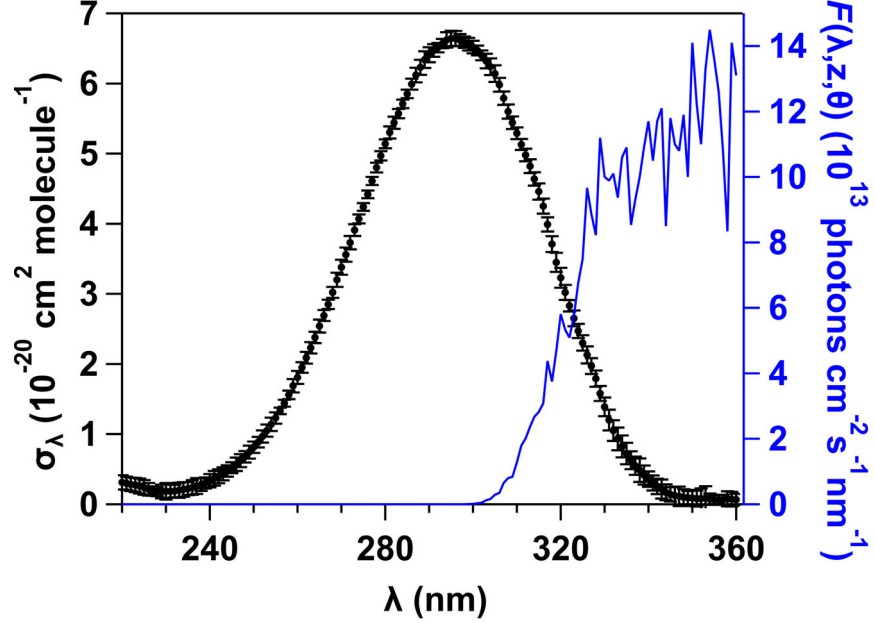

**Figure 1. UV absorption cross sections of 2MP at 298 K. The error bars represent the statistical uncertainty (±2σ).**

Photolysis of 2MP follows first order kinetics, where $J$ is the photolysis rate coefficient to be determined from the slope of $ln([2MP]_0/[2MP]_t)$ versus time plot. In Figure S3 the average values of individual $ln([2MP]_0/[2MP]_t)$ obtained in ten experiments, corrected with $k_{heterog}$, are plotted against $t$. Under the experimental irradiation conditions, $J$ (±2σ) was $(2.2\pm0.1)\times10^{-5}$ $s^{-1}$. The effective photolysis quantum yield of 2MP at $\lambda \geq 290$ nm, $\Phi_{eff}$(2MP), was calculated from Eq. (E3) as described in Asensio et al. (2022).

$$J \cong \Phi_{\text{eff}}(2MP) \sum_{290nm}^{360nm} I_\lambda \sigma_\lambda \Delta\lambda \qquad \text{(E3)}$$

where $I_\lambda$ (in photons cm$^{-2}$ s$^{-1}$ nm$^{-1}$) is the irradiance of the solar simulator at each wavelength, $\sigma_\lambda$ is the UV absorption cross section of 2MP measured in this work and $\Delta\lambda = 1$ nm. Considering all these parameters, the effective quantum yield of 2MP is $\Phi_{\text{eff}}(2MP) = (0.32 \pm 0.03)$. No quantum yields of 2MP have been previously reported. However, for structurally similar aldehydes like hexanal or 3-methylpentanal, the reported $\Phi_{\text{eff}}$ were similar (0.28±0.05 (Wenger, 2006) and 0.34 (Rebbert and

Ausloos, 1967), respectively) to that determined in this work.

The UV photodissociation of 2MP can proceed through the different channels (R1a-R1d) similar to other aldehydes (see *e.g.* Wenger (2006)). Under atmospheric conditions, the Norrish type I (R1a) and type II (R1c) processes are the most important pathways (Moortgat, 2001). R1a and R1d channels are radical-forming channels whilst the Norrish type II and R1b processes produce stable molecules with no radical formation (pentane and CO for the R1b process and 2-buten-2-ol and

propene for the R1c process). Radical reactions from R1a and R1d channels may generate 2-pentanone and 2-pentanol as primary products of the reaction of $CH_3CH_2CH_2CH(CH_3)$ and $CH_3CH_2CH_2CH(CH_3)CO$ with $O_2$, respectively.

$$CH_3CH_2CH_2CH(CH_3)C(O)H + h\nu \rightarrow CH_3CH_2CH_2CH(CH_3) + HCO \qquad \text{(R1a)}$$
$$\rightarrow CH_3CH_2CH_2CH_2CH_3 + CO \qquad \text{(R1b)}$$
$$\rightarrow CH_3(OH)C=CHCH_2 + CH_3CH=CH_2 \qquad \text{(R1c)}$$

$$\rightarrow CH_3CH_2CH_2CH(CH_3)CO + H \qquad \text{(R1d)}$$

In Figure 2, the IR spectra recorded before and after 150 min of irradiation are presented. The identified photolysis products were 2-pentanone (bands around 3000-2900 cm$^{-1}$ and 1732 cm$^{-1}$), which indicate that the Norrish type I and R1d channels are open, and CO (characteristic band 2000-2300 cm$^{-1}$), formed both directly *via* R1b and/or from the fast reaction of HCO radicals with $O_2$. However, it cannot be assured that the co-product of CO, pentane, was formed since the IR bands used for the

identification of pentane (3000-2800 cm$^{-1}$ and 1480-1340 cm$^{-1}$) can also be attributed to propene. Therefore, the Norrish type II process could also be open. The reference IR spectra used in the identification of these products are shown in Figure S4. After subtracting the IR features of CO, 2-pentanone, pentane, and propene, some IR features are still in the residual spectrum (see Figure 2c). The remaining features can be assigned to oxygenated compounds formed from the chemistry of the radicals generated in reactions R1a and R1d (for example, 2-pentanol from oxidation of $CH_3CH_2CH_2CH(CH_3)$ and

$CH_3CH_2CH_2CH(CH_3)CO$ radicals) or butanone formed by the keto-enolic tautomerism of 2-buten-2-ol (reaction product of R1c). The quantification of the reaction products identified was very imprecise. For that reason, no molar yield is provided.

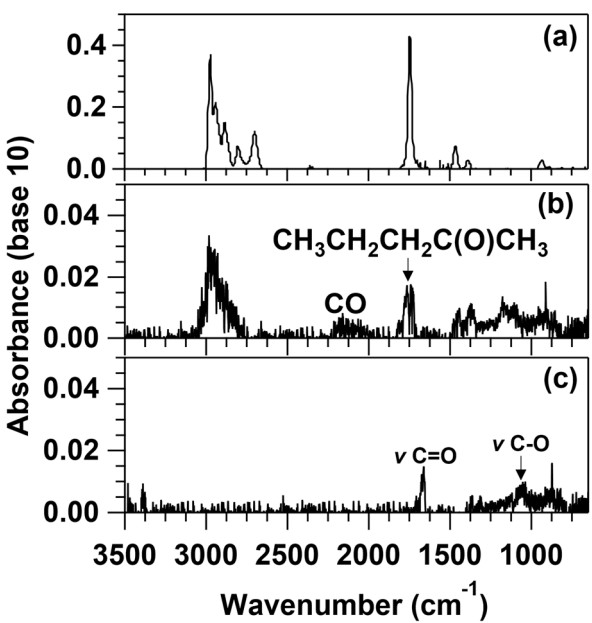

**Figure 2. FTIR spectra of the 2MP/air mixture. (a) before irradiation; (b) after 150 min of photolysis (features of unreacted 2MP were subtracted); and (c) residual spectrum after**

**subtraction of the reference spectra of the identified products (CO, 2-pentanone, pentane, and propene) shown in Figure S4.**

### 3.2 Kinetics and products of the 2MP+Cl reaction

Relative rate plots (Eq. (E2)) are shown in Figure S5. The slope of these plots yields the individual $k_{Cl}/k_{ref}$ values

listed in Table 1 for each reference compound. The good linearity of such plots suggests that the extent of secondary reactions was negligible. The average rate coefficient $k_{Cl}$, reported here for the first time, is:

$$k_{Cl} = (2.2\pm0.4)\times10^{-10} \text{ cm}^3 \text{ molecule}^{-1} \text{ s}^{-1}.$$

The stated uncertainty ($\pm2\sigma$) in $k_{Cl}$ includes the propagation of the reported errors in $k_{ref}$, the statistical errors from the slope of the plots shown in Figure S5, and the systematic uncertainties (estimated to be $\pm10\%$ of $k_{Cl}$). In the SI, a detailed description

of the error analysis can be found.

**Table 1**. Summary of relative rate measurements for 2MP at $298\pm2$ K and $760\pm5$ Torr of air. Errors include statistical $\pm2\sigma$ and 10% systematic uncertainties (see text).

| Reference | $k_{Cl}/k_{ref}$ | $k_{ref}$ ($10^{-10}$ cm$^3$ molecule$^{-1}$ s$^{-1}$) | $k_{Cl}$ ($10^{-10}$ cm$^3$ molecule$^{-1}$ s$^{-1}$) |
|---|---|---|---|
| Propene | $0.92\pm0.01$ | $2.23\pm0.30$[a] | $2.1\pm0.3$ |
| Isoprene | $0.55\pm0.01$ | $4.30\pm1.16$[b] | $2.4\pm0.6$ |
| | | | Average: **2.2±0.4** |

[a] Ceacero-Vega et al. (2009), [b] Orlando et al. (2003)

The gas chromatography-mass spectrometry technique was only used as an identification technique. Figure S6 shows the chromatogram obtained before ($t = 0$) and after 40 min of irradiation. The following reaction products were identified: acetaldehyde, butanedial, acetic acid, 2-pentanone, 3-pentanone, butanal, and 2-methylbutanoic acid. The mass spectra of 2MP and the identified products are shown in Figure S7. No products were observed during the UV light exposure of 2MP in the

absence of Cl-precursor or in its dark reaction with $Cl_2$.

Not all the reaction products identified by GC-MS were detected by FTIR spectroscopy. The identified products were CO, HCl, 2-pentanone, acetaldehyde, 2-methylbutanoic acid, acetic acid, and formaldehyde (Figure S8). The IR spectra used for quantification of 2-pentanone, acetic acid, acetaldehyde, formaldehyde, HCl, and CO were recorded in our lab, while those of pentane, 2-methylbutanoic acid, and propene were taken from the NIST FTIR database (Figure S4). After subtracting the

IR features of all these products, some bands between 1900 and 1100 cm$^{-1}$ still remain in the residual spectrum (see the bottom of Figure S8), that could be due to the other compounds, with lower concentration, identified by GC-MS, for which the detection by FTIR was difficult due to the lack of characteristic bands in the residual spectrum. The remaining bands could correspond to butanal, butanedial or a mixture of both. The molar yields for the major products, $Y_{Product}$, were obtained from the slope of [Product] versus consumed [2MP] plots (Figure S9a). $Y_{Product}$, ($\pm2\sigma$) were ($84.6\pm3.4$)% for HCl, ($23.9\pm0.6$)% for

2-pentanone, and ($11.1\pm0.3$)% for acetaldehyde.

All reaction products identified by GC-MS and FTIR spectroscopy, except CO, were also observed by PTR-ToF-MS. These products are formaldehyde ($CH_2OH^+$, m/z= 31.02), acetaldehyde ($C_2H_4OH^+$, m/z= 45.03), acetic acid ($C_2H_4O_2H^+$, m/z=61.02), 2-pentanone ($C_5H_{10}OH^+$, m/z=87.08), 2-methylbutanoic acid ($C_5H_{10}O_2H^+$, m/z=103.13), butanedial ($C_4H_6O_2H^+$, m/z= 87.04), butanal ($C_4H_8OH^+$, m/z = 73.06). In addition, other products were also observed such as methanol ($CH_4OH^+$,

265   m/z=33.03), methylglyoxal ($C_3H_4O_2H^+$, m/z=73.03), and propanoic acid ($C_3H_6O_2H^+$, m/z=75.04), but at lower concentrations. The PTR-ToF-MS signal from the identified products was calibrated to ensure the accuracy in the quantification. From the product yield plots shown in Figure S9b, $Y_{Product}$ ($\pm2\sigma$) were ($18.9\pm0.4$)% for 2-pentanone and ($14.1\pm1.1$)% for acetaldehyde.

The time evolution of 2MP and the products identified with an average ion concentration greater than $2\times10^{11}$ molecule cm$^{-3}$ (8.1 ppb) is shown in Figure 3. Although these products were also formed during the 2MP exposure to the UV light in the test experiments (without Cl$_2$) performed prior the Cl reaction, their concentration was negligible (<$4\times10^{10}$ molecule cm$^{-3}$ ≡ 1.6 ppb) compared with the observed levels after the Cl reaction. For minor products, no yield values are provided.

The presence of HCl as a primary product in the R2 reaction indicates that the reaction proceeds via H-abstraction mainly from one of the following three sites:

$$CH_3CH_2CH_2CH(CH_3)C(O)H + Cl \rightarrow CH_3CH_2CH_2CH(CH_3)C(O) + HCl \qquad (R2a)$$
$$\rightarrow CH_3CH_2CH_2C(CH_3)C(O)H + HCl \qquad (R2b)$$
$$\rightarrow CH_3CH_2CHCH(CH_3)C(O)H + HCl \qquad (R2c)$$

Two of the major products, HCl and acetaldehyde, can be explained by any of the three pathways mentioned above (R2a-c). The first step leads to the formation of HCl and a radical that, after several reactions with O$_2$ or RO$_2$ radicals, forms acetaldehyde. Reactions R2a and R2b lead to the generation of 2-pentanone, another major product. In reaction R2a, the carbonyl radical obtained after the H-abstraction from the -C(O)H group reacts with O$_2$ and RO$_2$ radicals yielding, after subsequent reactions, 2-pentanone. In addition, the decomposition of the radical prior to the formation of 2-pentanone generates butanal, detected by PTR-ToF-MS and GC-MS. Other minor products, such as formaldehyde and methanol, are generated through the three pathways as final decompositions products. Finally, the decomposition of CH$_3$CH$_2$CH$_2$C(CH$_3$)C(O)H and CH$_3$CH$_2$CHCH(CH$_3$)C(O)H radicals from the R2b and R2c reactions generate methylglyoxal, another minor product detected by PTR-ToF-MS.

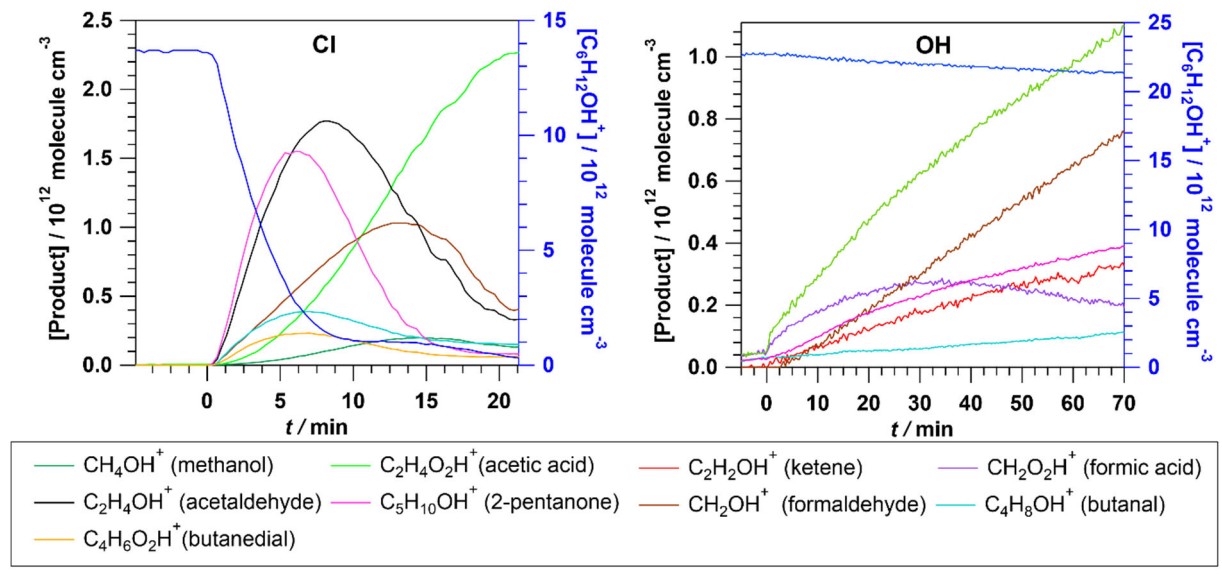

**Figure 3. Time evolution of 2MP (C$_6$H$_{12}$OH$^+$) and the products measured by PTR-ToF-MS during the Cl and OH reactions.**

### 3.2.3 Formation of Secondary Organic Aerosols (SOAs)

The size distribution of the particles formed in the Cl+2MP reaction with mobility diameters ($D_p$) between 6 and 523 nm is shown in terms of the normalized particle number, $dN/d\log D_p$, and mass, $dM/d\log D_p$, in Fig. 4. As shown in Fig. 4.a, after 2-4 min ($t$=0 min corresponds to the beginning of Cl reaction), the maximum $dN/d\log D_p$ corresponds to particles with a diameter of ca. 100-150 nm, while the maximum $dM/d\log D_p$ was reached at $t$ = 10 min for particles of ca. 500 nm (Fig. 4.b), close to the maximum $D_p$ that can be detected by the FMPS. Due to this instrumental limitation, the total SOA mass formed could not be inferred.

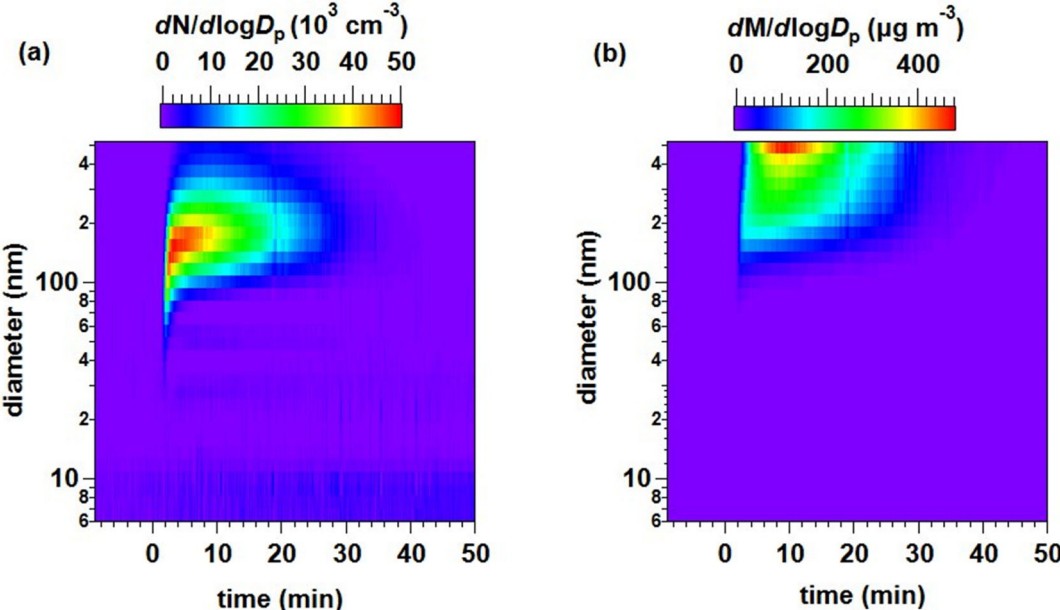

**Figure 4. Time evolution of the size of the SOA generated in the 2MP + Cl reaction in terms of the normalized particle number (a) and mass (b).**

### 3.3 Kinetics and products of the 2MP with OH reaction

Since no pressure dependence of the individual second-order rate coefficient, $k_{OH}(T)$, for the OH+2MP reaction was found between 50 and 500 Torr, as shown in Table S4, the rate coefficients at a single temperature were obtained by plotting $k'-k'_0$ $vs$ [2MP]$_0$ at different total pressures (Figure 5). For example, at room temperature:

$$k_{OH}(298 \text{ K}) = (3.3 \pm 0.3) \times 10^{-11} \text{ cm}^3 \text{ molecule}^{-1} \text{ s}^{-1}$$

which is in excellent agreement with that reported by D'Anna et al. (2001), $k_{OH}(298 \text{ K}) = (3.32 \pm 0.14) \times 10^{-11} \text{ cm}^3 \text{ molecule}^{-1} \text{ s}^{-1}$. To our knowledge, this is the first kinetic study of reaction R3 as a function of temperature. In the studied range of temperature (263–353 K), $k_{OH}(T)$ shows a slightly negative temperature dependence, increasing around 61% from 353 to 263 K, as shown in Table S4. The values of $k_{OH}(T)$ in the 263–353 K range fit to the Arrhenius equation (Figure S10). The pre-exponential

factor $A=(6.18 \pm 0.94) \times 10^{-12}$ cm$^3$ molecule$^{-1}$ s$^{-1}$ and $E_a/R=(509 \pm 45)$ K, i.e., an activation energy of $E_a=-(4.23 \pm 0.38)$ kJ mol$^{-1}$ was found. Uncertainties in $A$ and $E_a$ correspond to 2σ statistical errors.

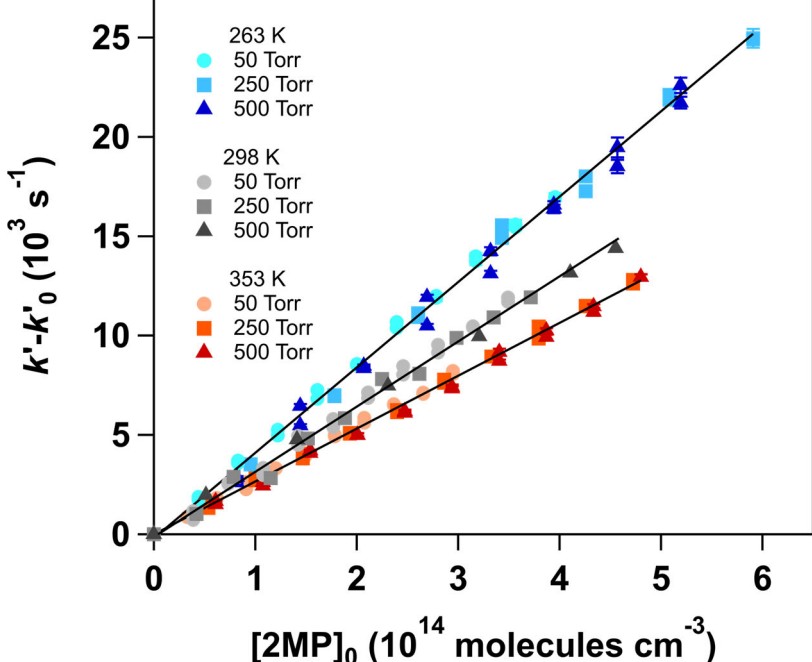

**Figure 5. Examples of the $k'–k'_0$ vs $[2MP]_0$ plots (Eq. ES3) at 263 K, 298 K and 353 K in the 50-500 Torr pressure range.**

Figure 3 shows the temporal profile of 2MP and the products identified in the OH-reaction by PTR-ToF-MS. In the timescale of the experiment, 2MP is less reactive towards OH radicals than towards Cl atoms, as shown in Figure 3. The major products were acetaldehyde, formaldehyde, and 2-pentanone with lower concentrations than those generated in the Cl-reaction. The reason of such observations can also be related to a low OH concentration generated in the photolysis of $H_2O_2$. This suggests that the OH reaction proceeds by a H-abstraction pathway similarly as the Cl reaction. Other minor products were ketene ($C_2H_2OH^+$, m/z=43.02), formic acid ($CH_2O_2H^+$, m/z=47.01), and butanal. Nevertheless, as confirmed in prior experiments in the absence of OH radicals, acetaldehyde, 2-pentanone, and formaldehyde were also formed, at lower concentrations than the observed after the OH reaction, in the dark reaction of 2MP with $H_2O_2$ ($1.6 \times 10^{11}$, $3.59 \times 10^{10}$ and $3.37 \times 10^{10}$ molecules $cm^{-3}$ (6, 1, and 1 ppb), respectively) and by UV photolysis ($3.76 \times 10^{11}$, $4.53 \times 10^{10}$, and $1.01 \times 10^{11}$ molecules $cm^{-3}$ (10, 2, and 4 ppb), respectively).

## 4 Atmospheric implications

Considering the most important degradation pathways (UV photolysis and reactions with Cl atoms and OH radicals), the lifetime of 2MP, $\tau_{2MP}$, can be estimated according to Eq. (E4).

$$\tau_{2MP} = \frac{1}{J(z,\theta)+k_{Cl}[Cl]+k_{OH}[OH]} \tag{E4}$$

where $J(z,\theta)$ is the photolysis rate of 2MP at a certain altitude (z) and solar zenith angle ($\theta$); $k_{Cl}$ and $k_{OH}$ are those determined in this work at 298 K; and [Cl] and [OH] are the concentrations of Cl atoms and OH radicals, which depend on the considered scenario (Table 2).

In this work, the following scenarios for a mid-latitude urban atmosphere were considered:

(i)   *Inland*: for a winter and a summer day at 13:00 h in Ciudad Real (z=0.6 km).

(ii)  *Coastal:* for a winter and a summer day in Valencia (z=0 km) at dawn, where the Cl chemistry can play a significant role.

**Table 2. Oxidant concentrations used in the estimation of the lifetime of 2MP.**

| Scenario | [OH] (radicals cm$^{-3}$) | Reference | [Cl] (atoms cm$^{-3}$) | Reference |
|---|---|---|---|---|
| $i$ | $10^6$ | Prinn et al. (2001) | $10^3$ | Singh et al. (1996) |
| $ii$ | $10^5$ | Holland et al. (2003) | $10^5$ | Spicer et al. (1998) |

The photolysis rate coefficient $J(z,\theta)$ was calculated as follows (Jiménez et al., 2007):

$$J(z,\theta) \cong \Phi_{\text{eff}}(2MP) \sum_{\lambda>290nm} F(\lambda,z,\theta)\sigma_\lambda \Delta\lambda \qquad \text{(E5)}$$

$\sigma_\lambda$ used in the calculation were those listed in Table S3 and $\Phi_{\text{eff}} = 0.32$, determined in this work. $F(\lambda,z,\theta)$ (in photons cm$^{-2}$ nm$^{-1}$ s$^{-1}$) is the solar spectral actinic flux for a specific $\theta$ in the troposphere, obtained using the TUV radiative transfer model (5.3 version) developed by Madronich and Flocke (1999) and $\Delta\lambda = 1$ nm. In Table S5, the photolysis rate coefficients obtained for scenarios $i$ and $ii$ are summarized and were used to calculate the lifetime of 2MP due to this process as $\tau_{hv} = 1/J(z,\theta)$ (see Table 3).

Considering the concentrations of oxidants provided in Table 2, the calculated lifetime of 2MP due to the reaction with OH radicals ($\tau_{OH}=1/\{k_{OH}[OH]\}$) is 8 hours in a mid-latitude inland urban atmosphere and 3 days in a coastal atmosphere at dawn. The lifetime of 2MP due to the Cl-reaction ($\tau_{Cl}=1/\{k_{Cl}[Cl]\}$) is 52 days in a mid-latitude inland urban atmosphere, but 12 hours in a mid-latitude coastal urban atmosphere at dawn. As shown in Table 3, the contribution of each process can be derived considering the individual lifetimes. In a mid-latitude inland urban atmosphere, the dominant degradation pathway for 2MP is the reaction with OH radicals in winter, while UV photolysis is a competing process in summer.

Tropospheric removal of 2MP by NO$_3$ radicals is negligible ($\tau_{NO3}$ = 41 hours) compared to the diurnal removal (on the order of few hours). To estimate $\tau_{NO3}$, the rate coefficient for the 2MP+NO$_3$ reaction reported by D'Anna *et al.* (2001) has been considered. In the estimation, a 24-h average [NO$_3$] is preferred, even though NO$_3$ is photolyzed at daytime. Based on a nighttime average [NO$_3$] of $5 \times 10^8$ radicals cm$^{-3}$ (Shu and Atkinson, 1995), the 24-h average [NO$_3$] considered here is $2.5 \times 10^8$ radicals cm$^{-3}$.

**Table 3. Estimated lifetimes of 2MP due to the investigated processes.**

| Scenario | $\tau_{hv}$ | $\tau_{OH}$ | $\tau_{Cl}$ | $\tau_{2MP}$ |
|---|---|---|---|---|
| $i$ | 21 hours [a]  8 hours [b] | 8 hours | 52 days | 6 hours [a]  4 hours [b] |
| $ii$ | 29 days [a]  3 days [b] | 3 days | 12 hours | 11 hours [a]  9 hours [b] |

[a] in wintertime; [b] in summertime

Regarding the impact of 2MP degradation on air quality, the consequences extend beyond the formation of carbonyl compounds mentioned earlier in low-NOx areas. The detected products, as summarized in Table 4, highlight the presence of various VOCs that, in polluted environments, may play a significant role in the generation of photochemical smog. For instance, 2-pentanone, which emerges as the most abundant VOC resulting from the Cl+2MP reaction and UV photolysis of 2MP, is also formed in the OH+2MP reaction. Additionally, acetaldehyde and formaldehyde, the predominant VOCs in the OH+2MP reaction, may further contribute to photochemical smog.

Table 4. Summary of identified reaction products in the reactions R1, R2, and R3 and analytical techniques used.

| Product | Chemical Formula | Process | | | Detection technique | | |
|---|---|---|---|---|---|---|---|
| | | 2MP + hν | 2MP + Cl | 2MP + OH | FTIR | PTR-ToF-MS | GC-MS |
| Hydrogen chloride | HCl | | Major | | Yes | | |
| Formaldehyde | $CH_2O$ | ✓ | | Major | Yes | Yes | |
| Formic acid | $CH_2O_2$ | | | ✓ | | Yes | |
| Methanol | $CH_4O$ | | ✓ | | | Yes | |
| Ketene | $C_2H_2O$ | | | ✓ | | Yes | |
| Acetaldehyde | $C_2H_4O$ | | ✓ | Major | Yes | Yes | Yes |
| Acetic acid | $C_2H_4O_2$ | | ✓ | | Yes | Yes | Yes |
| Propanoic acid | $C_3H_6O_2$ | | ✓ | | | Yes | |
| Methylglyoxal | $C_3H_4O_2$ | | ✓ | | | Yes | |
| Propene | $C_3H_6$ | ✓ | | | Yes | | |
| Butanal | $C_4H_8O$ | | ✓ | ✓ | | Yes | Yes |
| 2-Pentanone | $C_5H_{10}O$ | Major | Major | ✓ | Yes | Yes | Yes |
| Butanedial | $C_5H_8O_2$ | | ✓ | | Yes | Yes | Yes |
| Pentane | $C_5H_{12}$ | ✓ | | | Yes | | |
| 2-Methylbutanoic acid | $C_5H_{10}O_2$ | | ✓ | | Yes | Yes | Yes |

As 2MP is primarily emitted to the atmosphere from anthropogenic sources, $NO_x$ chemistry typical of polluted environments cannot be fully decoupled. When high levels of $NO_x$ are present, ozone is formed. The amount of tropospheric ozone that may be generated from the degradation of 2MP initiated by OH radicals (POCP, Photochemical ozone creation potential) is estimated to be 46 (relative to ethene, POCP = 1). This parameter for 2MP and, therefore, its contribution to photochemical smog, is on the same order of magnitude as other aldehydes (Jenkin et al., 2017). Moreover, in these environments, it is likely that toxic and irritant peroxyacyl nitrates (PANs) will be formed.

Furthermore, the evidence obtained from this research demonstrates the fast formation and subsequent growth of SOAs. Nonetheless, these findings must be interpreted cautiously, as the concentrations of $Cl_2$ and 2MP employed in the SOA experiments were notably higher in comparison to what is typically expected in the atmosphere.

In polluted sites, such as coastal areas or locations with ceramic industries and coal-fired power plants (Galán et al., 2002; Sarwar and Bhave, 2007; Deng et al., 2014), the photolysis of $Cl_2$ is the dominant daytime Cl atom source together with $ClNO_2$ (see Tham et al. (2016); Liu et al. (2017); Xia et al. (2020)). While the Cl+2MP reaction could eventually play a role in the atmospheric HCl budget, given the current 2MP emissions, it is likely to have a relatively minor impact on acid rain compared to other sources of HCl.

**Conclusions**

Once 2MP is emitted, it can be degraded in few hours during daytime (4-11 hours), depending on the location and season. The direct impact of the degradation of 2MP not only leads to the formation of harmful carbonyl compounds and the growth of submicron particles, exacerbating photochemical smog and atmospheric pollution, but it can also contribute to the atmospheric budget of HCl, especially in regions where the concentration of Cl atoms is high, although it will have a small impact on acid rain.

**Author contribution**

M. As., M. An., and S. B. conducted the experiments, and analyzed the experimental data. E.J. and J.A. designed and supervised the experiments and managed the project. All the co-authors have contributed to prepare the manuscript and to discuss the obtained results.

## Competing interests

The authors declare that they have no conflict of interest.

## Acknowledgements

This work has been supported by the regional government of Castilla-La Mancha and the European Regional Development Fund (FEDER) through the CINEMOL project (Ref.: SBPLY/19/180501/000052) and by the University of Castilla-La Mancha – UCLM (REF: 2021-GRIN-31279). María Asensio and Sergio Blázquez also acknowledge CINEMOL and UCLM (Plan Propio de Investigación), respectively, for funding their contracts during the performance of this investigation.

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
