# Peer review of "Atmospheric impact of 2-methylpentanal emissions: Kinetics, photochemistry, and formation of secondary pollutants"

_EGUsphere, 2023_

## Referee Comment (RC1)

**Review of "Atmospheric impact of 2-methylpentanal emissions: Kinetics, photochemistry, and formation of secondary pollutants" - egusphere-2023-1616**

This excellent paper details experimental studies of the decomposition 2-methylpentanal (2MP) in air. A variety of techniques were employed to determine photochemical parameters for 2MP photolysis, and reactions with OH and Cl-atoms. The atmospheric implications of these results have been assessed and lifetimes for the three breakdown pathways have been evaluated in different scenarios. I strongly recommend publication in ACP, subject to consideration of the following corrections / suggestions for improvement:

- Results, e.g. rate constants are presented to three significant figures throughout. Is there any justification for this degree of precision? The standard error values associated with many of these results would suggest not, e.g. line 193 (and also in table below), $k_{Cl}$ = (2.21 +/- 0.35), whereas conventionally this would be reported as (2.2 +/- 0.4) due to the size of the standard error.
- There is some discussion in section 4 around the contribution of 2MP and its breakdown products to photochemical smog formation. Jenkin and co-workers have presented a straightforward rubric whereby these effects may be quantified via calculation of POCP estimates. https://doi.org/10.1016/j.atmosenv.2017.05.024
  Such an analysis should be done in this work and would offer the opportunity to quantitatively compare the impacts of 2MP to similar compounds.
- Conclusions mention (line 331 – 332) contributions to acid rain via HCl formation. Whilst I appreciate that HCl was detected from the Cl + 2MP experiments, I cannot imagine a real-world scenario where emission of 2MP would make any difference to HCl budgets. Any Cl-atoms would be rapidly converted to HCl via other more common VOC, whether or not 2MP was emitted. Further, the following line "regions where Cl2 emissions are prevalent" is odd. There have been many reports of ClNO2 impacting on oxidant budgets via Cl-atom formation; direct Cl2 emissions would be uncommon across most of the globe. I suggest you remove these lines from the conclusions, or back-up assertions around HCl / Cl2 with explicit referencing.

**More trivial / typographical:**

Abstract – I am unconvinced it was helpful to detail acronyms in the abstract. FTIR, PLP-LIF and SOA are in such common usage that I would not bother with long form "Fourier Transform…" If this contravenes editorial policy, then write the long form but drop the acronym.

Introduction line 27 should be "particulate matter" not "the particulate matter"

Line 106 title 2.2.2 – italic "T" in $k(T)$

Experimental line 121 section 2.3 title is not very descriptive – suggest "Product studies of 2MP reactions with Cl (R2) and OH (R3)

Figure 3 caption – odd formatting of ion species, with several carbon "C" symbols out of alignment with the rest of the formula.

---

## Author Response (AR1)

**Author's Response Manuscript: egusphere-2023-1616**

**REPLY TO REFEREE #1**

We greatly appreciate the reviewer's comments on our manuscript which definitively will improve it. All typographical errors will be corrected in the revised version of the manuscript.

*Reviewer: Results, e.g. rate constants are presented to three significant figures throughout. Is there any justification for this degree of precision? The standard error values associated with many of these results would suggest not, e.g. line 193 (and also in table below), kCl = (2.21 +/- 0.35), whereas conventionally this would be reported as (2.2 +/- 0.4) due to the size of the standard error.*

> **Authors' reply**: In our previous works, we generally express the rate coefficients with three significant figures, as the literature rate coefficients for the used reference compounds. However, the referee is right that due to the size of the standard error, it would be better to reduce the number of significant figures. The significant figures in the rate coefficients will be reduced as suggested by the reviewer.

*Reviewer: There is some discussion in section 4 around the contribution of 2MP and its breakdown products to photochemical smog formation. Jenkin and co-workers have presented a straightforward rubric whereby these effects may be quantified via calculation of POCP estimates. Such an analysis should be done in this work and would offer the opportunity to quantitatively compare the impacts of 2MP to similar compounds.*

> **Authors' reply**: As we did in other papers (see Antiñolo et al., *J. Phys. Chem. A* 2017, 121, 8322−8331; Jiménez et al., *Environ. Sci. Technol.* 2016, 50, 1234−1242; or González et al. *Environ. Sci. Pollut. Res.* 2015, 22:4793–4805), the amount of tropospheric ozone that may be generated from the degradation of 2MP initiated by OH radicals has been included as suggested by the reviewer. POCP is estimated relative to that from ethene (POCP = 1) to be 46, which is on the same order of magnitude of POCPs for other aldehydes (Jenkin et al., *Atmos. Environ.* 2017, 163, 128-137). Based on the estimated POCP for 2MP, it can be concluded that its contribution to photochemical smog is similar to that for other aldehydes. This discussion will be included in the revised version of the manuscript.

*Reviewer: Conclusions mention (line 331 – 332) contributions to acid rain via HCl formation. Whilst I appreciate that HCl was detected from the Cl + 2MP experiments, I cannot imagine a real-world scenario where emission of 2MP would make any difference to HCl budgets. Any Cl-atoms would be rapidly converted to HCl via other more common VOC, whether or not 2MP was emitted. Further, the following line "regions where Cl$_2$ emissions are prevalent" is odd. There have been many reports of ClNO$_2$ impacting on oxidant budgets via Cl-*

*atom formation; direct $Cl_2$ emissions would be uncommon across most of the globe. I suggest you remove these lines from the conclusions, or back-up assertions around HCl / $Cl_2$ with explicit referencing.*

**Authors' reply**: Our aim was to highlight that HCl is among the detected products and can contribute to acidify the atmosphere. The sentence "...*it (2MP) can also contribute to the occurrence of acid rain, especially in regions where $Cl_2$ emissions are prevalent*" refers to the fact that photolysis of $Cl_2$ is the dominant daytime Cl atom source together with $ClNO_2$ (see Tham et al., 2016, *Atmos. Chem. Phys.*, 16, 14959–14977; Liu et al., 2017, *Environ. Sci. Technol.*, 51, 9588–9595; Xia et al., 2020, *Atmos. Chem. Phys.*, 20, 6147–6158) in polluted sites. The Cl+2MP reaction ultimately would contribute to the atmospheric budget of HCl. However, we agree with the referee that current 2MP emissions, much lower than other sources of HCl, would not significantly increase the atmospheric budget of HCl.
We will reword this part of the paper to make this statement clearer.

*Reviewer:* Abstract – I am unconvinced it was helpful to detail acronyms in the abstract. FTIR, PLP-LIF and SOA are in such common usage that I would not bother with long form "Fourier Transform…" If this contravenes editorial policy, then write the long form but drop the acronym.

**Authors' reply**: The instructions of the journal for submission indicate that the abstract should be intelligible to the general reader without reference to the text and that abbreviations should not be included without explanations. However, the stated acronyms are commonly used by the atmospheric chemistry community. Thus, we will keep the acronyms without explanations, as suggested by the referee.

**REPLY TO REFEREE #2**

We greatly appreciate the reviewer's comments on our manuscript which definitively will improve it.

All typographical errors will be corrected in the revised manuscript.

*Reviewer: Although the authors have chosen to report molecular number densities which are relevant for the kinetic data described, for ease of reading it would also be beneficial for mixing ratios to be included. For example, on line 80 the authors state that "initial 2MP ranged from 1.12 to 6.55 x $10^{16}$ molecules cm$^{-3}$" but the pressure was not explicitly stated in the main manuscript. The text stated "under atmospheric conditions" and it can be inferred that this was referencing lower tropospheric pressures but this should still be stated explicitly.*

**Authors' reply**: Certainly, in *atmospheric field measurements* pollutant concentrations are usually reported in terms of mixing ratios (ppm, ppb, ppt…). However, bimolecular rate coefficients of gas-phase reactions are commonly expressed in cm$^3$ molecule$^{-1}$ s$^{-1}$, that is the reason of expressing concentrations in molecules cm$^{-3}$. For the absolute kinetic measurements, we prefer to report all concentrations in terms of molecular number densities, but for the experiments performed in the presence of air (i.e., Cl-kinetics (298±2 K and 760±5 Torr) and photolysis experiments (298±1 K and 760±2 Torr) mixing ratios will also be included in the revised version of the manuscript, as suggested by the reviewer. Also, the temperature and pressure conditions of the so-called "atmospheric conditions" will be included in the revised manuscript.

*Reviewer: I find section 2.3 very unclear. A lot of beneficial information has been put into the SI, which makes reading this section quite laborious. I recommend that the authors re-write this section to include a more detailed description of their experiments, including key experiment conditions (e.g., including amounts of regent and oxidant). Furthermore, more details on the SOA experiments should be included.*

**Authors' reply**: As suggested by the referee more experimental details will be added in the revised manuscript, since multiple experiments using different instrumentation have been performed.

The following aspects will be moved from the SI to the main text: experimental details on the photodissociation of 2MP, the relative kinetic studies, and the product studies, both gaseous products and SOA.

*Reviewer: The methods section should include more details Section 2.4 should either be integrated into the text or moved to the SI.*

**Authors' reply**: As suggested by the referee more experimental details will be given the revised manuscript and section 2.4 will be moved to the SI.

*Reviewer: The organization of the SI would benefit from some work. The SI is comprised of additional experimental details and results, which are currently organized either individually or as part of a section. The authors should consider a table of contents to aid in locating content of interest.*

**Authors' reply**: It is a good idea to include a table of contents in the SI. The revised version of the manuscript will include it.

*Reviewer: The introduction would benefit from more specific details including emission rates of 2-methylpentanal, to help give readers more context on its significance. For example, the authors mention that 2MP can be emitted from "some foods and waste streams" as well as a number of other industrial sources, but then also mention that "2MP has been detected in ambient air at the foot of the Everest mountain". The former imply that 2MP mostly come from localized sources but the latter could imply that 2MP is both long-lived and well mixed throughout the troposphere, or has a specific local source in this remote region. The authors should provide more context, to better preface their photochemical and kinetic data. Furthermore, the introduction would benefit from more references to inland $Cl_2$ sources and how these could impact 2MP oxidation chemistry. $Cl_2$ is often emitted from power plants from the combustion of coal (e.g., Sarwar and Bhave (2007)) and is of greater global significance than ceramic industries.*

**Authors' reply**: Although there are some previous works that have detected 2-methylpentanal from different sources as stated in the original manuscript, unfortunately no emission rates were reported and the introduction of the manuscript cannot be improved in terms of 2MP emissions as suggested by the reviewer. However, we will provide more context in this section to better preface the photochemical and kinetic data presented in this work. As pointed out by the reviewer, another inland $Cl_2$ source very relevant globally is the emission from coal-fired power plants (Sarwar and Bhave, 2007, *J. Appl. Meteorol. Climatol.*, 46, 1009–1019; Deng et al., 2014, *Res. Environ. Sci.*, 27, 127–133). This source and the corresponding references will be added in the following sentence:

*"Globally, ceramic industries and coal-fired power plants are relevant sources of $Cl_2$ inlands (Galán et al., 2020; Sarwar and Bhave, 2007, Deng et al., 2014)."*

*Reviewer: The authors discuss the importance of 2MP originating primarily from anthropogenic sources, but then discuss doing experiments in a $NO_x$-free environment. Although it is important to probe this chemistry in the absence of $NO_x$ to determine the reaction kinetics and products of 2MP with individual oxidants, this chemistry cannot be fully decoupled. At a minimum, discussion regarding what impact the presence of $NO_x$ will have on these reactions and*

*product formation should be included in the discussion. Furthermore, the authors discuss in part the diurnal importance of Cl and OH chemistry but do not mention NO₃. Although the reaction of NO₃ for 2MP has been studied (D'Anna et al 2001, Table 1), and is much slower this should still be discussed briefly. This reaction should also be included in the mechanisms listed in lines 49 to 51.*

**Authors' reply**: The reviewer is right that $NO_x$ chemistry is important in polluted environments where most anthropogenic emissions take place. This chemistry wasn't originally included in the manuscript since the experiments in this work were performed in a $NO_x$-free environment. In polluted atmospheres, where $NO_x$ is present, it is likely that toxic and irritant peroxyacyl nitrates (PANs) will be formed from the reactions studied in this work together with the products already mentioned in the original manuscript. This will be included in the atmospheric implications section of the revised manuscript.

Regarding the discussion on the contribution of the $2MP+NO_3$ reaction to the overall loss of 2MP, note that the aim of this work is focused on its diurnal atmospheric chemistry and that is the reason why $NO_3$ reaction was not mentioned in the manuscript. The fact that we focused on the diurnal chemistry will be included in the introduction of the manuscript to avoid any confusion. Furthermore, the reactions in lines 49-51 (of the original manuscript) are those studied in this work, so we don't believe that $2MP + NO_3$ reaction should be included in the introduction, since the study of the kinetics or products of this reaction was not part of the objectives of this work. Nevertheless, a discussion about the $NO_3$ chemistry will be included in the atmospheric implications section as suggested by the reviewer. In order to compare the tropospheric lifetime of 2MP due to $NO_3$-reaction ($\tau_{NO_3}$) with that for the OH-reaction a 24-h average $[NO_3]$ is preferred, even though $NO_3$ is photolyzed at daytime. Based on a 12-h average $[NO_3]$ of $5 \times 10^8$ radicals $cm^{-3}$ (Shu and Atkinson. *J. Geophys. Res. Atmos.*, 100, 1995, 7275-7281), the 24-h average $[NO_3]$ was considered $2.5 \times 10^8$ radicals $cm^{-3}$. To estimate $\tau_{NO_3}$ the rate coefficient for the $2MP+ NO_3$ reaction reported by D'Anna et al. (2001, *Phys. Chem. Chem. Phys.*, 3, 3057-3063) was considered. The conclusion is that the $NO_3$ chemistry of 2MP is negligible ($\tau_{NO_3}$ = 41 hours) compared to the diurnal removal (between 4 and 11 hours in the scenarios presented in this work).

*Reviewer: The uncertainties associated with the determined photochemical and kinetic data (e.g., rate coefficients) could be further clarified. In particular it would be beneficial for the authors to include instrumental uncertainties, since the rate coefficients for the Cl and OH experiments were determined using different methods. The authors should consider adding this as a section in the SI.*

**Authors' reply**: The reported uncertainties in the rate coefficients *k* and *J* were obtained from the statistical analysis. For the Cl-reaction experiments it also includes the error propagation considering the

uncertainty in $k_{ref}$. Instrumental/systematic uncertainties were not included originally, but they do not exceed ±10% of the determined parameter. The total uncertainties in $k_{Cl}$, $k_{OH}$, and $J$ that will be stated in the revised manuscript, $\Delta k_{2MP}$ and $\Delta J$, will be calculated considering the systematic uncertainties, as follows:

$$\Delta k_{2MP}= \sqrt{\Delta k_{2MP} \ (stat)^2 + \Delta k_{2MP} \ (syst)^2}$$

$$\Delta J= \sqrt{\Delta J \ (stat)^2 + \Delta J \ (syst)^2}$$

Thus, the uncertainties in $k_{Cl}$ ($\Delta k_{Cl}$), in $k_{OH}$ at room temperature ($\Delta k_{OH}$), and in $J$ ($\Delta J$) are:

$$\Delta k_{Cl} = \sqrt{(0.4\times10^{-10})^2 + (0.2\times10^{-10})^2} = 0.4\times10^{-10} \ cm^3 \ molecule^{-1} \ s^{-1}$$

$$\Delta k_{OH} = \sqrt{(0.1\times10^{-11})^2 + (0.3\times10^{-11})^2} = 0.3\times10^{-11} \ cm^3 \ molecule^{-1} \ s^{-1}$$

$$\Delta J = \sqrt{(0.1\times10^{-5})^2 + (0.2\times10^{-5})^2} = 0.2\times10^{-5} \ s^{-1}$$

*Reviewer: Implication of 2MP oxidation on aerosol formation. The concentrations that experiments were conducted at, for both 2MP and $Cl_2$, were very high ($\times10^{14}$ molecules $cm^{-3}$ or 10s of ppm at 1 atm). These mixing ratios would not be atmospherically relevant in terms of SOA formation. The rapid appearance of a mode just below 200 nm suggests rapid nucleation and condensation of oxidized VOCs. As such, the use of these data to justify the ability of 2MP to form SOA is questionable, especially given the unclear experimental description. Additionally, with the current experimental description, it is unclear whether these experiments were conducted in the presence of isoprene/ propene, which would impact the formation of SOA. The authors should clarify these details. Furthermore, was no SOA observed for reactions with OH? The authors mention that fewer products were formed under OH oxidation that they believe was due to low OH concentrations. Did the authors conduct the SOA experiments with OH as well or just with Cl?*

**Authors' reply**: We agree with the reviewer that the concentrations of $Cl_2$ and 2MP used in the SOA experiments were quite high compared to the presumable atmospheric levels. The reason for this is an experimental limitation: if lower concentrations had been used, it would have been difficult to observe changes in the 2MP concentrations by

our FTIR system. We will point out in the revised manuscript that the reported results should be considered with caution.

SOA experiments were performed in the absence of isoprene or propene. No seeds were added to the gas mixture. This information will be included in the revised manuscript.

*"SOA experiments were performed in the absence of the reference compounds used in the relative kinetic study (isoprene or propene) to avoid any interference from their degradation initiated by Cl atoms. No seed was added to the gas mixture either."*

SOA experiments were performed only for the 2MP + Cl reaction in this work. In the conducted OH experiments, the aim was not to determine SOA formation given that the reaction mechanism for the two oxidants is quite similar (observed products were almost the same) and an analogous behaviour is expected.

*Reviewer: Line 89-90- The authors state that "the heterogeneous loss process accounts for 39% of the total 2MP loss." Have the authors done any experiments to determine whether 2MP could be driven off the chamber walls during photolysis? And if not, can the authors comment on how they believe this would impact the calculation of the photolysis rate coefficient, J. Furthermore, it would be beneficial to include a summary of these experiments in the SI.*

**Authors' reply**: Wall deposition of 2MP and further desorption into the gas phase are processes influenced by temperature and not by light itself. The heterogeneous loss experiments were performed in a thermostated cell at 298 K to avoid the heating of the cell when the light from the solar simulator hit it. An increase of the temperature cell, when light was on, causes an increase on the concentration of the reactant and a decrease on $J$. The summary of these experiments was already included in the SI although, as pointed out above, it will be moved to the main manuscript in the revised version.

*Reviewer: Line 97- Did the authors mean reference compounds? For simplicity perhaps state the reference compound in the text. It is unclear if only one or both were used.*

**Authors' reply**: Yes, FTIR spectroscopy was used for detecting 2MP and both reference compounds, not only one. Each Cl-kinetic experiment was conducted in the presence of only one reference compound. Then, experiments were repeated using another reference compound. The text in the manuscript will be rewritten to avoid any confusion.

*"The mixture of 2MP, $Cl_2$, one of the reference compounds, and synthetic air was introduced into the 16-L cell described above and FTIR spectroscopy was used as detection technique to monitor 2MP and the reference compounds (isoprene or propene)."*

*Reviewer:* Line 131- Should give at a minimum an approximate concentration for these species in the text. Otherwise, it is very vague. Furthermore, it would be beneficial for the authors to comment on the approximate concentration of radicals (OH and Cl) generated for these experiments.

**Authors' reply**: As suggested by the Reviewer, the estimated radical concentrations will be commented in the description of the experimental details in section 2.3 of the revised manuscript. Concentrations of molecular species will also be embedded in the text. For that reason, Table S2 will be removed from the SI in the revised version.

Once $k_{OH}$ and $k_{Cl}$ were determined in the kinetic experiments, stationary OH and Cl concentrations can be estimated using the first order loss rate of 2MP measured ($ln\{[2MP]_0/[2MP]_t\} = k$ [oxidant] $t$) by FTIR and PTR-ToF-MS in the product studies and the experimental rate coefficient ($k = k_{OH}$ or $k_{Cl}$). In the figure on next page, an example of the linearized rate equation is presented for both detection methods. From the slope of such plots [OH] or [Cl], which are listed in the table below, are determined. For example, for the Cl-reaction, the [Cl]/[Cl$_2$]$_0$ ratios obtained are $1.03 \times 10^{-9}$ (FTIR, 3 actinic lamps) and $2.03 \times 10^{-6}$ (PTR-ToF-MS, 8 actinic lamps). As 2MP was not quantified by GC-MS, [Cl] is estimated from [Cl$_2$]$_0$ and considering the [Cl]/[Cl$_2$]$_0$ ratio determined in SOA experiments in which the same lamps were used to generate Cl.

| Detection Technique | $[H_2O_2]_0$ $(10^{14}$ cm$^{-3})$ [a] | **[OH][a] $(10^5$ cm$^{-3})$ [a]** | $[Cl_2]_0$ $(10^{14}$ cm$^{-3})$ [a] | **[Cl][a] $(10^6$ cm$^{-3})$ [a]** |
|---|---|---|---|---|
| FTIR | -
 - | -
 - | 1.6 - 5.4
 7.8 [b] | **2.0 – 3.1**
 **0.80 [b]** |
| GC-MS | - | - | 1.1 - 2.5 | **0.11 – 0.26** |
| PTR-ToF-MS | 3.8 | **4.8** | 0.11 - 0.14 | **10 – 22** |

[a] Estimated concentrations; [b] In SOA experiments, [Cl]/[Cl$_2$]$_0$ is $1.03 \times 10^{-9}$.

**FTIR**

[Figure]

$y = 1.77E\text{-}04x - 2.65E\text{-}02$
$R^2 = 9.95E\text{-}01$

[Figure]

**PTR-ToF-MS**

$y = 5.07E\text{-}03x - 5.68E\text{-}02$
$R^2 = 9.97E\text{-}01$

Y-axis: $\ln([2MP]_0/[2MP])_t$
X-axis: $t$ (s)

**Reviewer:** *Line 210-212- The authors state that there were still bands around 1900 and 1100 cm$^{-1}$ that remained in the residual spectrum that were attributed to lower concentration compounds that were identified by GC-MS. Given the retention times and mass spectra data of these compounds can the authors identify what they are? Please elaborate.*

**Authors' reply**: According to the mass spectra shown in Figure S7 of the original SI file, butanedial (HC(O)CH$_2$CH$_2$C(O)H) and butanal (CH$_3$CH$_2$CH$_2$C(O)H) were formed as products. This was confirmed by PTR-ToF-MS in which the peaks corresponding to C$_4$H$_6$O$_2$H$^+$ and C$_4$H$_8$OH$^+$ were observed. Therefore, the bands that remain in the residual FTIR spectrum correspond possibly to butanedial (no IR spectrum was available), butanal (see figure below) or a mixture of both. The following sentence will be included in the revised manuscript:

*"The remaining bands could correspond to butanal, butanedial or a mixture of both."*

[Figure]

**Reviewer:** *Line 290-292- Although the locations selected are fine to use as case studies, the authors should consider reframing the context in which they describe the Cuidad Real site as "a global atmosphere" and Valencia as a "local atmosphere" since these are very generic. The authors should consider describing the Cuidad Real site as a "mid-latitude inland urban atmosphere" and*

*the Valencia site as a "mid-latitude coastal urban atmosphere". Furthermore, the authors should be careful about not over interpreting these data on a global scale, and were as focus their discussion on implications in similar areas.*

> **Authors' reply**: We agree with the reviewer that describing Ciudad Real as a "global scale" location can be too ambitious. We will change the notation of "global scale" in the manuscript to better describe the location evaluated in this work.

*Reviewer: Line 324- I am skeptical about the impact this chemistry will have on the formation of acid rain. In regions with industrial sources, the emissions of $SO_x$ and $NO_x$ will still be the largest contributor to this process. The authors should consider removing this statement or adding additional references/ data to support this statement.*

> **Authors' reply**: Our aim was to highlight that HCl is among the detected products and can contribute to acidify the atmosphere. The sentence *"...it (2MP) can also contribute to the occurrence of acid rain, especially in regions where $Cl_2$ emissions are prevalent"* refers to the fact that photolysis of $Cl_2$ is the dominant daytime Cl atom source together with $ClNO_2$ (see Tham et al., 2016, *Atmos. Chem. Phys.*, 16, 14959–14977; Liu et al., 2017, *Environ. Sci. Technol.*, 51, 9588–9595; Xia et al., 2020, *Atmos. Chem. Phys.*, 20, 6147–6158) in polluted sites. The Cl+2MP reaction ultimately would contribute to the atmospheric budget of HCl. However, we agree with the referee that current 2MP emissions, much lower than other sources of HCl, would not significantly increase the atmospheric budget of HCl.
>
> We will reword this part of the paper to make this statement clearer.
>
> *"While the Cl+2MP reaction could eventually play a role in the atmospheric HCl budget, given the current 2MP emissions, it is likely to have a relatively minor impact on acid rain compared to other sources of HCl."*

*Reviewer: Table 4. 2MP and its "detection techniques" should be included in this summary table. It would also be beneficial to include the instrument LOD/ precision for these techniques. This would also help to provide context in terms of the uncertainties of the determined reaction rates, which are currently unclear.*

> **Authors' reply**: Obviously, 2MP was detected by all the analytical methods included in Table 4, otherwise we could not have used them to monitor the loss of 2MP and derive the rate coefficients and product yields. We think it is obvious that 2MP has to be monitored in the kinetics and products studies.
>
> The LOD precision of the analytical techniques will be preferably included in the experimental section and not in the revised Table 4, since no quantification is presented in this table. The LOD of the three techniques are:

$$LOD_{PTR-ToF-MS} = 1.2 \times 10^8 \text{ molecules cm}^{-3} \text{ (5 ppt)}$$

$$LOD_{GC-MS} = 1 \times 10^{11} \text{ molecules cm}^{-3} \text{ (4 ppb)}$$

$$LOD_{FT-IR} = 8 \times 10^{11} \text{ molecules cm}^{-3} \text{ (32 ppb)}$$

As for the concern of the Reviewer about the uncertainties of the determined reaction rate coefficients, it is clear that the concentrations used in the kinetic experiments are several orders of magnitude higher than the LOD of the techniques, so the uncertainties on the concentrations will be very low.

**Reviewer:** *SI Line 99- The authors state "concentrations were lower due to its high sensitivity". Do the authors mean that the detectable concentrations were lower due to the better instrument sensitivity or that they performed experiments at lower concentrations? According to Table S2 it is the former. Please clarify the text.*

**Authors' reply**: Yes, we mean that the detectable concentrations of PTR-ToF-MS were lower due to the better instrument sensitivity. The PTR-ToF-MS sensitivity is higher than the FTIR of SPME-GC/MS sensitivities, so it was possible to conduct experiments at lower concentrations, that are more similar to a real atmosphere. Furthermore, the PTR-ToF-MS would have got saturated if concentrations as high as the used for the other techniques were used.

**Reviewer:** *Figure S10- The authors give an example for experimental data conducted at 263 and 353 K, which represent the range of temperatures that kinetic experiments were conducted under. Since the authors are reporting that these are the first kinetic measurements for this reaction as a function of temperature, why not include data for 298 K which is of greater interest to the data being discussed in the main manuscript and also most readers.*

**Authors' reply**: The kinetic data at 298 K has been included in former Figure S10 and has been moved to the main text of the revised manuscript (now Figure 5).

[Figure]

**Figure 5.** Examples of the *k'–k'₀ vs* [2MP]₀ plots (Eq. ES3) at 263 K, 298 K and 353 K in the 50-500 Torr pressure range.

---

## Author Response (AR2)

**Authors' Response to Editor's Comments**
**Manuscript: egusphere-2023-1616**

We greatly appreciate the editor's comments on our manuscript.

***Editor:*** *I propose to undo the revisions in the abstract made to only show acronyms for FTIR, PLP-LIF and SOA. What is common to some people is uncommon to other. I would suggest using the full unabbreviated expressions.*

> **Authors' reply**: In the final manuscript, we have reverted the changes made on the acronyms in the abstract to comply with the journal guidelines.

***Editor:*** *In figure 1, words in the title of the right vertical axis do not all fall on one line. The same issue is in the title of the horizontal axis in Figure 5, as well as the formula of the molecule shown in Figure 2(b)*

> **Authors' reply**: The misalignment of some characters in the graphs was due to a problem when converting our MS Word to a pdf file. We have solved it by changing the format of our figures in the final manuscript.

***Editor:*** *I presume the absorbance in Figure 2 is calculated using log (base 10), not ln (base). It maybe worth specifying this.*

> **Authors' reply**: We have added to Figure 2 the fact that absorbance is reported as base 10.

***Editor:*** *The paper refers to particle diameters. It is customary to specify the type of diameter. I presume it is a mobility-equivalent diameter since the authors used an FMPS. I would explicitly state this.*

> **Authors' reply**: We have included in the text that $D_p$ corresponds to mobility diameters.